# Identification of Molecular Mechanisms in Radiation Cystitis: Insights from RNA Sequencing

**DOI:** 10.3390/ijms25052632

**Published:** 2024-02-23

**Authors:** Sabrina Mota, Elijah P. Ward, Sarah N. Bartolone, Michael B. Chancellor, Bernadette M. M. Zwaans

**Affiliations:** 1Department of Urology, William Beaumont University Hospital, Corewell Health System, Royal Oak, MI 48073, USA; 2Department of Urology, Oakland University William Beaumont School of Medicine, Rochester Hills, MI 48309, USA

**Keywords:** radiation cystitis, bladder, RNA sequencing, radiation therapy

## Abstract

Pelvic cancer survivors who were treated with radiation therapy are at risk for developing (hemorrhagic) radiation cystitis (RC) many years after completion of radiation therapy. Patients with RC suffer from lower urinary tract symptoms, including frequency, nocturia, pelvic pain, and incontinence. In advanced stages, hematuria can occur, potentially escalating to life-threatening levels. Current therapeutic options for RC are limited, partly due to ethical concerns regarding bladder biopsy in patients with fragile bladder tissue. This study aimed to leverage our established preclinical model to elucidate the molecular pathways implicated in radiation-induced tissue changes in the bladder. Female C57Bl/6 mice received a single dose of 40 Gy using CT-guided imaging and a two-beam irradiation approach using the SARRP irradiator. Bladders from irradiated and age-matched littermate controls were harvested at 1 week [n = 5/group] or 6 months [n = 5/group] after irradiation, RNA was harvested, and mRNA sequencing was performed at paired-end 150bp on the Illumina NovaSeq6000 with a target of 30 million reads per sample. Following RNA sequencing, thorough bioinformatics analysis was performed using iPathwayGuide v2012 (ADVAITA Bioinformatics). Findings of the RNA sequencing were validated using qPCR analysis. At 1 week post-irradiation, altered gene expression was detected in genes involved in DNA damage response, apoptosis, and transcriptional regulation. By 6 months post-irradiation, significant changes in gene expression were observed in inflammation, collagen catabolism, and vascular health. Affected pathways included the p53, JAK-STAT, and PI3K-Akt pathways. These findings were validated in vivo in bladder tissues from our preclinical model. This is the first study to determine the molecular changes in the bladder in response to radiation treatment. We have successfully pinpointed several pathways and specific genes that undergo modification, thereby contributing to the progression of radiation cystitis. These insights enhance our understanding of the pathophysiology of radiation cystitis and may ultimately pave the way to the identification of potential new therapeutic targets.

## 1. Introduction

Radiation therapy is a widely used treatment modality for pelvic cancers such as prostate, cervical, bladder, and colorectal cancer [1,2,3,4]. Despite its therapeutic effectiveness, radiation therapy can harm adjacent tissues such as in hemorrhagic radiation cystitis (RC), which is a tremendously challenging condition to treat and represents a health economic concern [5,6]. RC is a chronic, often irreversible bladder condition characterized by bladder fibrosis and vascular damage. Tissue remodeling, neovascularization, and angiogenesis coincides with RC symptoms such as frequency, nocturia, pelvic pain, incontinence, and hematuria, which can range from microscopic to gross hematuria with blood clots [7]. The presence of blood clots can cause bladder obstruction and block urine outflow. Bladder obstruction is an urgent medical condition and is primarily addressed through invasive medical procedures. Consequentially, upon RC diagnosis, arresting the bleeding is the primary focus of treatment. The severity of hematuria determines the course of action, with rest and hydration being the most conservative approach and formalin instillations or cystectomy being last resort measures [8,9].

The incidence rate of RC is challenging to determine due to its long latency and subsequent loss-to-follow-up of patients. However, it is estimated to be around 11.1–16.2% with a mean onset of 79.1 months in prostate cancer survivors and 3–6.7% after 1 year in cervical cancer survivors [10]. A study by Makino and collaborators reported a cumulative 5-year incidence of RC of 16.2% in prostate cancer survivors [11].

While treatments with limited effectiveness exist, there is no known cure for RC. RC is often diagnosed at a late stage when severe damage makes effective treatment difficult. Therefore, gaining a deeper understanding of the mechanisms underlying bladder fibrosis and vascular damage is crucial for early diagnosis and treatment. Our research group previously developed a preclinical model for radiation cystitis using CT-guided imaging and a 2-beam approach to deliver radiation to mouse bladders [12,13]. In this study, we utilized our established animal model of RC and performed RNA sequencing on bladder tissue to explore molecular pathways and gene expression changes induced by radiation. We analyzed multiple timepoints post-irradiation to gain valuable insights into the progression of RC.

## 2. Results

### 2.1. RNA-Seq Quality Control

The raw sequencing data had indicators of good quality sequencing including normal duplication rates, passing GC content, and a minimum of 30 million uniquely mapped reads. Screening for other species indicated that most samples primarily mapped to only the mouse genome. Diagnostic plots indicated that the data were consistent with the set of differentially expressed genes, including within-group variability.

### 2.2. Differential Expression Analysis

Differential expression (DE) analysis revealed a combined 40,025 unique genes for 1 week (17,893 genes) and 6 months post-IRR (22,132 genes) versus age-matched control bladders (Table 1).

Significant differentially expressed (SDE) genes were identified considering a defined experimental threshold of 0.05 for statistical significance (*p*-value) and a log of fold change overexpression with an absolute value of at least 0.585. For 1 week post-IRR (post radiation treatment) versus age-matched control samples, a total of 246 SDE genes were identified (Appendix A), and 607 genes for the 6 months post-IRR time point were identified (Figure 1 and Appendix A).

The iPathwayGuide scoring system identified 50 significant pathways for 1 week post-IRR (Appendix A) and 53 for 6 months post-IRR (Appendix A). For this analysis, pathway *p*-values are calculated based on the impact analysis method, considering over-representation of differentially expressed genes and the perturbation computed across the pathway topology. The 1 week comparison showed increased significance for p53 signaling, apoptosis, and cancer pathways, while the 6 month comparison showed increased inflammation, vascular health, and tissue remodeling pathways (Figure 2A). For downstream signaling pathways, Pi3k-Akt was significantly increased at 1 week and 6 months. JAK-STAT and Ras showed higher significance at 1 week post-RR, while NF-kappa B and HIF-1 showed higher significance at 6 months post-IRR (Figure 2B).

For gene ontology (GO) analysis, iPathwayGuide compares the number of DE genes for a specific biological process with the probability that DE genes occur naturally [14,15]. Top scoring biological processes identified by this analysis also revealed a significant increase in the intrinsic apoptotic signaling pathway mediated by p53 at 1 week post-IRR (7 differentially expressed genes out of 44 genes, Figure 3A). For the 6 month analysis, immune system process (155 differentially expressed genes out of 2371 genes, Figure 3B) and collagen catabolism process (8 differentially expressed genes out of 34 genes, Figure 3C) were significantly elevated (Table 2 and Table 3).

Comparison of SDE cytokine and chemokine pathways (Figure 4) reveals genes correlated with cell injury and apoptosis at 1 week post-IRR: growth differentiation factor 15 (Gdf15, related to response to cell injury); tumor necrosis factor receptor superfamily member 10b, (Tnfrsf10b, involved in positive regulation of apoptosis), ectodysplasin A2 receptor (Eda2r, involved in intrinsic signaling of an apoptotic pathway), and Fas (TNF receptor superfamily member 6, involved in extrinsic signaling of an apoptotic pathway). At 6 months post-IRR, there is an increase in pro-inflammatory SDE genes such as interleukin 27 (Il27), interleukin 18 (Il18), interleukin 12b (Il12b), and chemokine (C-C motif) receptor 7 (CCR7)—negative regulation of apoptotic process—Cxcr2.

Finally, we used RT-qPCR analysis to validate RNA sequencing results with respect to distinct pathways at 1 week and 6 months post-IRR (Figure 5). Overall, the trends from RNA sequencing data were validated by the qPCR analysis, though changes in gene expression levels compared to age-matched controls were less pronounced with qPCR. At 1 week post-IRR, key genes in apoptosis and the p53 pathway that were elevated included MDM2, BAX, PUMA, FAS, and Dr5 (Figure 5A). At 6 months, matrix metalloproteases that were elevated included MMP 3, 7, 8, 10, 12, and 27 (Figure 5B).

## 3. Discussion

Patients who are prone to developing radiation cystitis after pelvic radiation therapy exhibit distinct disease phases. Initially, there is an acute phase that lasts several weeks, followed by a symptom-free latent phase that can last from months to years. Eventually, a chronic and irreversible phase is observed that can take up to 15 years after radiation therapy to become apparent. The acute and chronic phases share common lower urinary tract symptoms, including urinary frequency, nocturia, and dysuria. However, the chronic phase is particularly characterized by varying degrees of hematuria, ranging from microscopic to severe gross hematuria with blood clots. In cases of chronic RC, the focus of treatment shifts to managing hematuria and preventing complications like bladder obstruction [8].

Our research group has previously developed a preclinical radiation model in which we have recapitulated the early and late voiding dysfunction phases of radiation cystitis. [12]. We have demonstrated that symptoms of acute RC histologically coincide with thinning and compromised barrier function of the urothelial wall and that the associated symptoms are resolved when the urothelial wall is restored. Symptoms during chronic RC, on the other hand, are associated with tissue fibrosis, loss of detrusor muscle, and vascular damage/remodeling [13]. Utilizing this model, the current study aimed to identify differentially expressed genes during the early and late stages of RC to better understand the condition’s progression and identify potential therapeutic targets.

### 3.1. Gene Expression Changes Post-Radiation

One week after irradiation, the biological processes that are primarily impacted are related to cell apoptosis and cell death. This is indicated by the activation of the intrinsic p53 apoptotic signaling pathway (GO:0072332) as well as positive regulation of release of cytochrome c from mitochondria (GO:0090200), which also falls under the regulation of the cysteine-type endopeptidases activity involved in the apoptotic process (GO:0043281) (Table 2). The p53 pathway is essential for maintaining genomic stability. In the event of DNA damage, p53 will arrest the cell cycle at the G1/S transition point. From here, the cell may undergo DNA repair, apoptosis, or senescence. Cytochrome c release is an early trigger of apoptosis and leads to caspase activation, typically caused by the permeabilization of the outer membrane of the mitochondria by proapoptotic stimuli. Previous studies have demonstrated that radiation results in thinning of the urothelium; thus, these findings may indicate that radiation causes significant DNA or cell damage leading to rapid loss of cells through apoptosis [7,13,16,17].

Six months post-radiation, while p53 signaling remained altered, apoptosis pathways were not prominently affected (Figure 2A). In the p53 pathway, cyclin-dependent kinase inhibitor 1A (P21) is one of the highest SDE genes at 1 week (LogFC = 2.04 and *p*-value = 0.000001) and 6 months after irradiation (LogFC = 1.37 and *p*-value = 0.000001), Appendix A. However, at 6 months, reprimo (Rprm) leads the SDE genes (LogFC = 1.82 and *p*-value = 0.000001). This gene is associated with regulation of mitotic cell cycle arrest at the G2 phase [18]. This suggests that the elevated p53 pathway activity may indicate a higher number of cells that are in cell cycle arrest for DNA repair or that have undergone cellular senescence. In fact, previous studies have demonstrated that irradiation can lead to premature cellular senescence [19].

At the six-month mark, the predominant changes are in collagen remodeling (GO:0030574) and immune system processes (GO:0002376). This is evidenced by the altered genes including chemokines and cytokines (i.e., IL-27, IL-18, CCL8, CCL22, and CXCL16) as well as other immune regulators (i.e., Gdf3, Fgr, and Ebi3), reflecting the chronic inflammatory state of the bladder (Figure 4). In addition, an array of matrix metalloproteinases is upregulated, indicating the active tissue remodeling that is taking place within the bladder (Figure 5). MMP3, for example, has been shown to be increased in idiopathic pulmonary fibrosis, and deletion of MMP3 protects mice from bleomycin-induced pulmonary fibrosis [20]. Likewise, MMP7 mediates renal fibrosis via B-catenin signaling, and serum MMP7 is elevated in patients with pulmonary fibrosis [21,22]. However, the role of MMPs in mediating tissue fibrosis may be tissue- and context-dependent. While MMPs 7, 8, 12, and 13 have all been described as having pro-fibrotic properties in lung and liver, they have also been shown to possess anti-fibrotic properties. As such, the role of the different MMPs and their interplay in radiation-induced bladder fibrosis require further investigation. Finally, pyroptosis (GO:0070269), the pro-inflammatory process of programmed cell death, is elevated at 6 months post-IRR (Table 3). Pyroptosis is a caspase-1-dependent cell death subroutine that is associated with the generation of pyrogenic mediators such as IL-1beta and IL-18, potentially contributing to cytokine upregulation.

### 3.2. Study Limitations and Future Directions

A limitation of our study is that RNA sequencing was performed on whole-bladder tissue, which may mask gene expression changes in specific, less abundant cell types like endothelial or basal urothelial cells. Future studies could include spatial transcriptomics or single-cell sequencing to identify cell-type-specific radiation-induced changes. In addition, this study was performed using mouse bladder tissue, and not all findings may be directly translated to humans. This study was conducted with a relatively small sample size (n = 5/group), and given the variability in response to irradiation, additional pathways may have been missed. Furthermore, with the introduction of image-guided intensity-modulated radiation therapy (IG-IMRT) in clinical practice, the precision with which radiation is delivered to its target is much improved over the years, which may result in a reduction in long-term damage to neighboring tissues. Nevertheless, our model closely mimics the human condition with respect to how external beam radiation is delivered, the multiple disease states, the symptoms, and the long latency of chronic RC. The identified pathways have a high degree of homology between mouse and humans, and thus we are confident the findings of this study will have significant relevance to human patients. Future studies could focus on validating altered pathways in a larger cohort or clinical samples as well as assessing which pathways drive the progression of RC. In addition, the therapeutic potential of identified pathways will require thorough investigation.

## 4. Materials and Methods

### 4.1. Radiation Cystitis Pre-Clinical Model

This study was performed with full approval from the Beaumont Institutional Animal Care and Use Committee (AL-20-04), in compliance with the NIH Guide for the Care and Use of Laboratory Animals. Animals were housed, treated, and cared for in an AAALAC-accredited facility. Female, 8-week-old C57Bl/6 mice were purchased from Charles River (Wilmington, MA, USA). C57Bl/6 mice were chosen as we have previously demonstrated that these mice are more susceptible to radiation-induced fibrosis than other strains [12]. Mice were randomly assigned to an irradiated or controlled treatment group [n = 5/group]. Radiation was delivered to the mouse bladder using the Small Animal Radiation Research Platform (SARRP) and a two-beam approach as previously described [12]. In brief, for radiation treatment, anesthesia was induced using 2.5–3% isoflurane through inhalation and maintained at 1.5–2% throughout the procedure (30 min). Mice were placed on the SARRP platform, and a CT image was taken to identify the bladder and position beams. All care was taken to avoid the spinal cord, the long bones, colon, and overlap of entrance and exit beams to help minimize damage to other organs such as the skin. Mice received a single dose of 40 Gy, evenly divided over the two beams, using a 5 × 5 mm collimator. This dose was chosen based on previous work in developing this radiation cystitis model [12,13]. The time points were chosen to reflect changes associated with acute (1 week) and chronic (6 months) RC. Mouse breathing and heart rate were continuously monitored during the procedure. After radiation treatment, mice were placed in a heated recovery cage and returned to regular housing when fully recovered, where they received mash and hydrogel for 7 days. Untreated mice were anesthetized for the same duration of time as their irradiated littermate controls. Bladders were harvested and snap frozen at 1 week and 6 months post-irradiation to represent acute and chronic RC, respectively.

### 4.2. RNA Isolation

All supplies were placed under a UV light for 15 min and wiped down with Invitrogen RNAse Zap (ThermoFisher Scientific, Waltham, MA, USA) prior to tissue processing. In total, 1 mL TRIzol and a ½ scoop of 0.9–2.0 mm stainless steel beads (Next Advance, Troy, NY, USA) were added to Eppendorf Safe-Lock tubes. Frozen bladders were added to tubes and placed in a Bullet Blender for 5 min at speed 10 at 4 °C. The mixture was then moved to a biosafety cabinet and incubated at room temperature for 5 min. Next, 200 µL chloroform was added to each tube containing bladder homogenate, incubated for 2 min at RT, then centrifuged at 12,000× *g* at 4 °C for 15 min. The top [aqueous] phase was carefully pipetted off, then added to a new pre-labeled tube; 500 µL 100% isopropanol was added and incubated for 10 min at RT and centrifuged at 12,000× *g* at 4 °C for 10 min. The supernatant was carefully removed and discarded, and 1 mL of 75% ethanol was added to gently re-suspend the pellet. The pellet was vortexed briefly then centrifuged at 7500× *g* at 4 °C for 5 min. The supernatant was then discarded, and the pellet allowed to air-dry for 5 min. Finally, the pellet was re-suspended in 20 µL of RNase-free H_2_O and incubated at 60 °C for 10 min at RT. All extracted RNA was assessed for quantity and quality using a Nanodrop 2000 (ThermoFisher) before downstream use. RNA was placed at −80 °C for long term storage or left on ice for immediate downstream applications.

### 4.3. RNA Sequencing, Quality Control, and Sequence Alignment

RNA sequencing was performed at the University of Michigan Advanced Genomics Core. Sequencing was performed with libraries constructed and subsequently subjected to 151 paired-end or single-end cycles on a NovaSeq-6000 platform (Illumina, San Diego, CA, USA). Adapters were trimmed from the read files using Cutadapt (v2.3). FastQC (v0.11.8) was used to ensure the quality of the data. Reads were mapped to the reference genome GRCm38 (ENSEMBL) using STAR (v2.6.1b) and assigned count estimates to genes with RSEM (v1.3.1). Alignment options followed ENCODE standards for RNA sequencing. FastQC was used in an additional post-alignment step to ensure that only high-quality data were used for expression quantification and differential expression.

### 4.4. Differential Expression Analysis

Data were pre-filtered to remove genes with less than 10 counts across all samples. Differential gene expression analysis was performed using DESeq2, using a negative binomial generalized linear model (thresholds: linear fold change >1.5 or <−1.5, Benjamini–Hochberg FDR Padj < 0.05). Plots were generated using variations of DESeq2 plotting functions and other packages with R (v3.3.3). Genes were annotated with NCBI Entrez GeneIDs and text descriptions. Functional analysis, including candidate pathways activated or inhibited in comparison(s), *p*-values, and GO-term enrichments, was performed using the iPathwayGuide scoring system (iPathwayGuide v2012; ADVAITA Bioinformatics, Ann Arbor, MI, USA). For GO Analysis, iPathwayGuide calculates two different types of pruning methods to decrease data redundance: high-specificity pruning and smallest common denominator pruning. The high-specificity pruning method identifies the most specific GO terms that are significantly associated with the set of DE genes. The smallest common denominator pruning method identifies the GO terms that best summarize the set of DE genes. Pathway topology was obtained from the KEGG database available at https://www.genome.jp/kegg/ (accessed on 5 May 2021) [23,24], and gene descriptions were obtained from NCBI available at https://www.ncbi.nlm.nih.gov/gene (accessed on 5 May 2021) [25].

### 4.5. Reverse Transcription of RNA

RNA concentrations were measured with a nanodrop (Thermo Fischer Scientific, Waltham, MA, USA) and subsequently diluted to 100 ng/µL from the initial concentration. All RNA samples had a purity >1.85. In total, 10 µL of diluted RNA was subsequently added to a PCR tube containing 2.0 µL 10× buffer, 2.0 µL 10× random primers, 0.8 µL 25× dNTPs, 4.2 µL H_2_O, and 1.0 µL reverse transcriptase (Omniscript RT Kit, Qiagen, Germantown, MD, USA). The reaction cycled 10 min at 25 °C, 2 h at 37 °C, and 5 min at 85 °C. The resultant cDNA was then analyzed via qPCR.

### 4.6. Quantitative PCR Analysis

In total, 10.0 µL of SYBR Select Master Mix (Thermo Fisher Scientific, Waltham, MA, USA), 0.6 µL of 10 µM forward primer, 0.6 µL of 10 µM reverse primer (Table 4), and 7.8 µL dH_2_O was added to each well of a 96-well MicroAmp Optical qPCR plate (Thermo Scientific). Additionally, 1.0 µL of cDNA was added to each well. dH_2_O was used as the non-template control. The PCR reaction was run under the following protocol: 50 °C for 2 min, 95 °C for 10 min, and (95 °C for 15 s, 62 °C for 1 min) for a total of 40 cycles. Three biological replicates were performed for each timepoint. The results were analyzed in QuantStudio (Applied Biosystems, Waltham, MA, USA), expression was normalized to 18 s, and fold increases were calculated between control and irradiated samples.

## 5. Conclusions

In summary, this study has delineated distinct gene expression patterns in acute versus chronic RC, with cell apoptosis and differentiation pathways being upregulated during the acute phase and tissue remodeling and immune response pathways upregulated during the chronic phase. Our findings underscore that radiation therapy induces long-term gene expression changes, shedding light on the chronic and progressive nature of RC. This understanding is crucial for developing targeted therapies and improving patient outcomes in radiation cystitis. Future studies should focus on validating the study findings in preclinical or clinical samples and assessing the role the identified pathways play in driving the progression of RC.

## Figures and Tables

**Figure 1 ijms-25-02632-f001:**
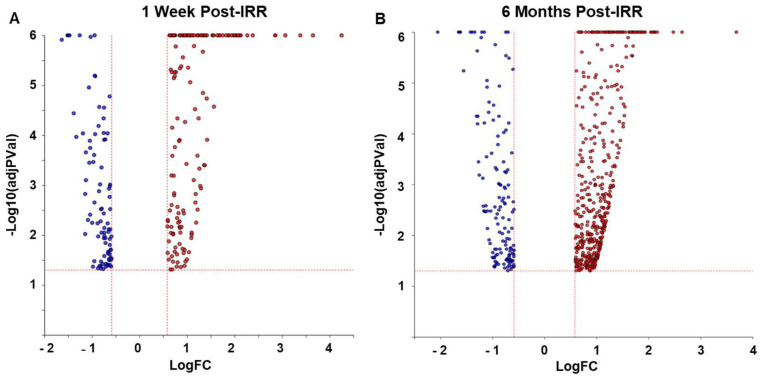
Volcano plot showing differentially expressed (DE) genes at (**A**) 1 week and (**B**) 6 months after irradiation treatment (post-IRR). The x-axis represents the measured expression fold change (log of the fold change expression), and the y-axis represents significance of the change (negative log_10_ of the *p*-value). The up-regulated genes are shown in red, while the down-regulated genes are blue. Red line defines threshold areas (Threshold *p* < 0.05 and logFC ≥ |0.585|).

**Figure 2 ijms-25-02632-f002:**
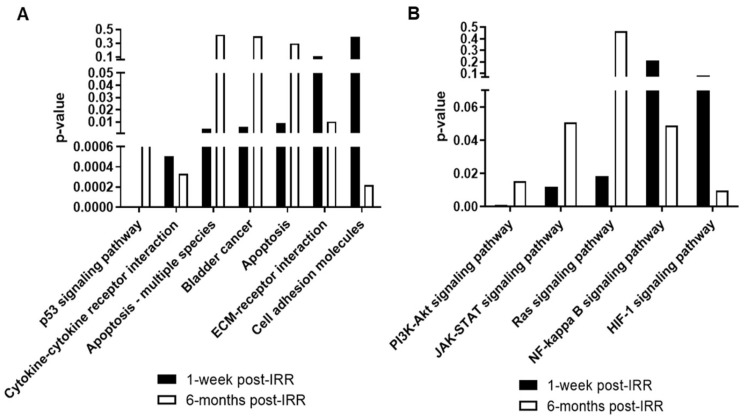
Comparison of relevant (**A**) biological pathways and (**B**) downstream signaling pathways at 1 week and 6 months versus time-matched control samples (N = 5; pathway from the KEGG database).

**Figure 3 ijms-25-02632-f003:**
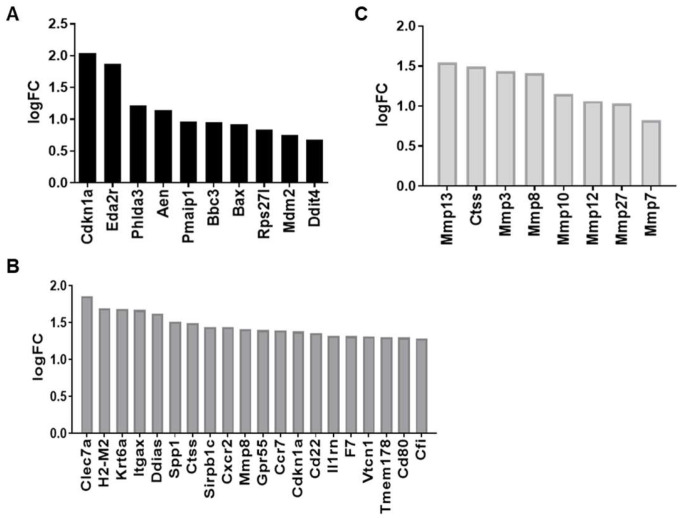
Significantly differentially expressed genes annotated to (**A**) intrinsic apoptotic signaling pathway by p53 class mediator immune system process at 1 week post-IRR, (**B**) immune system process at 6 months post-IRR, and (**C**) collagen catabolism process at 6 months post-IRR (N = 5). Immune system process showing only the top 20 out of 155 SDE genes. Gene symbols from (https://www.ncbi.nlm.nih.gov/gene; accessed on 5 May 2021), GO terms from (https://amigo.geneontology.org/amigo; accessed on 5 May 2021).

**Figure 4 ijms-25-02632-f004:**
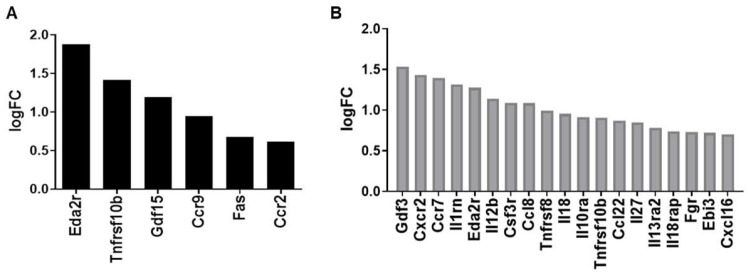
Comparison of significantly differentially expressed cytokine and chemokine pathways at (**A**) 1 week and (**B**) 6 months post-IRR compared time-matched control samples (N = 5). Ranked based on their absolute value of log fold change. Gene symbols from (https://www.ncbi.nlm.nih.gov/gene; accessed 5 May 2021).

**Figure 5 ijms-25-02632-f005:**
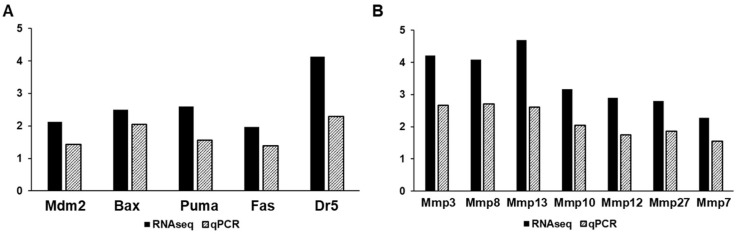
Comparison of fold change in GOI between RNA-seq and qPCR at (**A**) 1 week post-IRR and (**B**) 6 months post-IRR.

**Table 1 ijms-25-02632-t001:** Summary of RNA sequencing data including total genes, differentially expressed genes, significant pathways, and gene ontology (GO) at 1 week and 6 months post-irradiation versus age-matched control bladders, identified using iPathwayGuide scores. * Significant differentially expressed genes if *p* < 0.05 and logFC ≥ |0.585|.

	1 Week Post-IRR	6 Months Post-IRR
Total Measured Genes	17,893	22,132
Differentially Expressed Genes *	246	607
Significant Pathways	50	53
Gene Ontology (GO)	1510	1741

**Table 2 ijms-25-02632-t002:** Top scored biological processes at 1 week post-IRR identified by GO analysis using different types of pruning methods to reduce data redundance. Scores and pruning analyses were calculated using iPathwayGuide v2012 (ADVAITA Bioinformatics, Ann Arbor, MI, USA). Only the top 5 biological processes are listed for each pruning analysis.

1 Week Post-IRR vs. Age-Matched Control					
Pruning Type: None		Pruning Type: High-Specificity		Pruning Type: Smallest Common Denominator	
GO	*p*-Value	GO	*p*-Value	GO	*p*-Value
Ion transport	2.60 × 10^−10^	Intrinsic apoptotic signaling pathway in response to DNA damage by p53 class mediator	1.54 × 10^−2^	Intrinsic apoptotic signaling pathway by p53 class mediator	1.33 × 10^−3^
Signaling	6.90 × 10^−8^	Positive regulation of release of cytochrome c from mitochondria	9.14 × 10^−2^	Regulation of cysteine-type endopeptidase activity involved in apoptotic process	1.33 × 10^−3^
Regulation of cell motility	1.60 × 10^−7^	Endocardial cell differentiation	9.14 × 10^−2^	Positive regulation of release of cytochrome c from mitochondria	6.38 × 10^−2^
Intrinsic apoptotic signaling pathway by p53 class mediator	2.00 × 10^−7^	Negative regulation of fibroblast proliferation	9.14 × 10^−2^	Endocardial cell differentiation	7.11 × 10^−2^
Regulation of cellular component movement	3.40 × 10^−7^	Blood vessel remodeling	9.14 × 10^−2^	Negative regulation of fibroblast proliferation	7.62 × 10^−2^

GO: gene ontology.

**Table 3 ijms-25-02632-t003:** Top scored biological processes at 6 months post-IRR identified by GO analysis using different types of pruning methods to reduce data redundance. Scores and pruning analyses calculated using iPathwayGuide v 2012 (ADVAITA Bioinformatics, Ann Arbor, MI, USA). Only the top 5 biological processes are listed for each pruning analysis.

6 Months Post-IRR vs Age-Matched Ctrl					
Pruning Type: None		Pruning Type: High-Specificity		Pruning Type: Smallest Common Denominator	
GO	*p*-Value	GO	*p*-Value	GO	*p*-Value
Immune system process	6.50 × 10^−21^	Collagen catabolic process	2.08 × 10^−2^	Response to another organism	2.05 × 10^−5^
Response to external stimulus	1.70 × 10^−18^	Pyroptosis	2.08 × 10^−2^	Pyroptosis	6.24 × 10^−3^
Immune response	7.30 × 10^−17^	Negative regulation of mast cell degranulation	2.41 × 10^−2^	Collagen catabolic process	6.24 × 10^−3^
Response to another organism	3.20 × 10^−16^	Antigen processing and presentation of exogenous peptide antigen via MHC class II	2.41 × 10^−2^	Positive regulation of cytokine production	6.24 × 10^−3^
Response to external biotic stimulus	3.40 × 10^−16^	Positive regulation of dendritic cell antigen processing and presentation	2.66 × 10^−2^	B-cell-mediated immunity	6.24 × 10^−3^

GO: gene ontology.

**Table 4 ijms-25-02632-t004:** Primer sequences for RT-qPCR.

Gene	Forward Primer (5′-3′)	Reverse Primer (5′-3′)
*Mmp3*	GGACCAGGGATTAATGGAGATG	TGAGCAGCAACCAGGAATAG
*Mmp8*	TCAACCAGGCCAAGGTATTG	ATGAGCAGCCACGAGAAATAG
*Mmp13*	GACACAGCAAGCCAGAATAAAG	GGAAAGCAGAGAGGGATTAACA
*Mmp10*	GGTGTGGTGTTCCTGATGTT	CCTGTAGGTGATGTGGGATTTC
*Mmp12*	GACATCTTGGCTCCCTATCTTC	TGGACAATACACCAGTCAGTTT
*Mmp27*	AGAGGATTCCCACAGAGGATAA	CCTCGACCCATGGAAGAAATAG
*Mmp7*	GAGTGCCAGATGTTGCAGAATA	ATCCACTACGATCCGAGGTAAG
*Mdm2*	AGCTGACAGAGAATGATGCTAAA	GGAAGTCGATGGTTGGGAATAG
*Noxa1*	GCCATCCTCATCTACGTGTAAG	TCCAGATGCCAGCAAACTATC
*Bax*	GGAGATGAACTGGACAGCAATA	GAAGTTGCCATCAGCAAACAT
*Puma*	CTGGAGGGTCATGTACAATCTC	ACCTAGTTGGGCTCCATTTC
*Fas*	CCTCCAGTCGTGAAACCATAC	TCTTGCCCTCCTTGATGTTATT
*Dr5*	GATGGAGCCAGGAAGATCAAG	AGTCAGCTCTCAGCAAGTTTAG
*18s*	CCGCAGCTAGGAATAATGGA	CGGTCCAAGAATTTCACCTC

## Data Availability

Data is contained within the article or Appendix A.

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
