# Peer review of "Identification of Molecular Mechanisms in Radiation Cystitis: Insights from RNA Sequencing"

_ijms, 2024, doi:10.3390/ijms25052632_

Round 1

Reviewer 1 Report

Comments and Suggestions for Authors

The authors presented in the manuscript very interesting study conducted on mice to show the biological effects on adjacent organs to prostate after radiotherapy, since about 16% of patients develop long term effects of radiation cystitis and there is no known cure for them.

They used global differential profiling of bladder tissue from healthy mice which were not irradiated and tissue one week and 6-months after irradiation. The analysis of the changes in signaling pathways allowed them to identify 50 significantly  changed pathways pointing out the biological processes which were affected after irradiation.

The authors critically described also the limitations of the study. I would add also the information about IMRT – new technology of radiotherapy. Intensity-modulated radiation therapy (IMRT) , which has revolutionized radiation oncology in recent years, is expected to reduce radiation exposure to periprostatic organs, especially bladder and rectum.

5-year cumulative incidence of RC was 16,2%, consistent with the reported incidence of RC, the 5-eyar cumulative incidence of severe RC was only 3% and no case required urinary divertion. This low incidence of severe RC could be explained by the introduction of IMRT. With the advent of dose-escalating radiation therapy, it was presumed that radiation doses to target organs have increased, while decreasing to adjacent organs, resulting in less toxicity.

They used global differential profiling of bladder tissue from healthy mice which were not irradiated and tissue one week and 6-months after irradiation. The analysis of the changes in signaling pathways allowed them to identify 50 significantly  changed pathways pointing out the biological processes which were affected after irradiation.

Author Response

Thank you for bringing this to our attention. We have added a sentence to the study limitations stating that IG-IMRT may result in reduction of long-term damage to neighboring tissues.

Reviewer 2 Report

Comments and Suggestions for Authors

The article reports the results of a study that used RNA sequencing to analyze the gene expression changes in the bladder of mice after radiation treatment, which can cause a chronic condition called radiation cystitis (RC). The study found that different pathways and genes were affected at 1 week and 6 months post-irradiation, reflecting the acute and chronic phases of RC. The study also identified potential therapeutic targets for RC based on the molecular insights. Specific comments:

1.          The abstract provides a good overview of the background, aims, methods, results, and implications of the study. However, it could be improved by adding some specific details, such as the number of mice used, the name of the bioinformatics tool used for functional analysis, and the potential therapeutic targets identified.

2.          The materials and methods section describes the experimental procedures in a clear and detailed manner. However, it could be improved by providing some additional information, such as the source and characteristics of the mice used, the rationale for choosing the dose and time points of irradiation, the quality control metrics and thresholds for RNA sequencing, and the statistical methods and software used for data analysis.

3.          Some of the figures are not labeled properly or are too small to read. I recommend revising these figures to make them more clear and readable.

4.          The discussion section interprets the results in the context of the existing literature and highlights the novel and significant aspects of the study. However, it could be improved by addressing some limitations or challenges of the study, such as the sample size, the variability, the reproducibility, or the generalizability of the findings. You could also discuss some future directions or implications of the study, such as the validation of the potential therapeutic targets in vivo or in clinical trials, or the comparison of the molecular mechanisms of RC with other types of bladder disorders.

5.          The conclusions section summarizes the main findings and implications of the study. However, it could be improved by providing a more balanced and critical evaluation of the study. For example, you could discuss the strengths and weaknesses of the study, the consistency or discrepancy of the results with the previous studies, the potential impact or application of the findings in the clinical practice or the future research, and the unanswered questions or the next steps of the study.

6.           The funding section discloses the sources of financial support for the study. However, it could be improved by providing some additional information, such as the grant number, the name of the funding agency, the country of the funding agency, and the role of the funding agency in the study. You could also declare if there was no specific funding for this study.

7.          The conflicts of interest section declares any financial or non-financial interests that could influence the conduct or reporting of the study. However, it could be improved by providing some additional information, such as the nature of the conflict, the amount of the financial interest, the duration of the conflict, and the management of the conflict. You could also declare if there were no conflicts of interest in this study.

Author Response

Thank you for your thorough review. We have addressed your concerns to the best of our ability. 

  1. We have included additional detail in the abstract as requested. We did not include any potential therapeutic targets as we do not feel it is correct to speculate on potential therapeutic targets without having more concrete evidence. Instead, we have softened the last sentence.
  2. We have added information about the source and characteristics of the mice and rationale for using dose and time points. For qPCR, we assessed sample purity prior to cDNA synthesis and we included a no-temp control. RNAseq thresholds were set as linear fold change >1.5 or <-1.5, and Benjamini-Hochberg FDR Padj <0.05. The software used for statistical analysis was ADVAITA Bioinformatics Software. We have provided as much information on RNAseq as provided to us by the core facility.
  3. All figures have been modified to improve clarity.
  4. The study limitations section has been expanded and we have added possible future avenues based on the findings described in this study.
  5. We have added additional information to the conclusion as requested.
  6. An updated funding statement has been added to the cover page. 
  7. An updated conflict of interest statement has been added to the cover page.

Reviewer 3 Report

Comments and Suggestions for Authors

The current study aimed to explore molecular pathways and gene expression changes induced by radiation in animal model of radiation cystitis.

The topic sounds very attractive due to the the current scenario. Radiation therapy, indeed, was established therapeutic care both for bladder cancer (PMID= 35625979 ) and for other malignancies, such as prostate and colorectal (PMID= 37995359, 31176433, 21057123). This concept should be stressed in the current manuscript. Indeed, the prevalence of the RC is increasing during the last decades, representing an health economic concern (especially due to the hyperbaric treatment PMID: 19484683). Specifically, the methodology which the authors  relied on was a case-control study where RNA sequencing on bladder tissue was performed.

The  follow-up to establish the mutations burden was settle at 1-week and 6 months after the irradiations. In my opinion, it should be added also a follow up at 3 months. Are the mutations burden increase after 3 mo threshold? Or was there virtually the same compared to 6-mo?

The results obtained were very interesting. At 6 mo indeed, the authors observed mutations in collagen and immune system molecules. This result definitely could have a pivotal role in the future research on target therapy.  

Author Response

We appreciate your thoughtful comments on our manuscript. We have added the provided references to emphasize the therapeutic importance of RT in various pelvic cancers.

We had also included a 3-months' timepoint in our study. The findings of the 3-months' timepoint were virtually the same as the 6-months' time point.